# Ferric Chloride Controls Citrus Anthracnose by Inducing the Autophagy Activity of *Colletotrichum gloeosporioides*

**DOI:** 10.3390/jof9020230

**Published:** 2023-02-09

**Authors:** Yuqing Wang, Xiaoxiao Wu, Yongqing Lu, Huimin Fu, Shuqi Liu, Juan Zhao, Chaoan Long

**Affiliations:** 1National Key Laboratory for Germplasm Innovation & Utilization of Horticultural Crops, Wuhan 430070, China; 2Key Laboratory of Horticultural Plant Biology of Ministry of Education, Wuhan 430070, China; 3National R&D Center for Citrus Preservation, Wuhan 430070, China; 4National Centre of Citrus Breeding, Wuhan 430070, China; 5College of Horticulture & Forestry Sciences of Huazhong Agricultural University, Wuhan 430070, China; 6Guangxi Laboratory of Germplasm Innovation and Utilization of Specialty Commercial Crops in North Guangxi, Guilin 541004, China; 7Guangxi Citrus Breeding and Cultivation Research Center of Engineering Technology, Guilin 541004, China; 8Guangxi Academy of Specialty Crops, Guilin 541004, China

**Keywords:** citrus, *Colletotrichum gloeosporioides*, ferric chloride, autophagy, ROS, membrane integrity

## Abstract

*Colletotrichum gloeosporioides* causes citrus anthracnose, which seriously endangers the pre-harvest production and post-harvest storage of citrus due to its devastating effects on fruit quality, shelf life, and profits. However, although some chemical agents have been proven to effectively control this plant disease, little to no efforts have been made to identify effective and safe anti-anthracnose alternatives. Therefore, this study assessed and verified the inhibitory effect of ferric chloride (FeCl_3_) against *C. gloeosporioides*. Our findings demonstrated that FeCl_3_ could effectively inhibit *C. gloeosporioides* spore germination. After FeCl_3_ treatment, the germination rate of the spores in the minimum inhibitory concentration (MIC) and minimum fungicidal concentration (MFC) groups decreased by 84.04% and 89.0%, respectively. Additionally, FeCl_3_ could effectively inhibit the pathogenicity of *C. gloeosporioides* in vivo. Optical microscopy (OM) and scanning electron microscopy (SEM) analyses demonstrated the occurrence of wrinkled and atrophic mycelia. Moreover, FeCl_3_ induced autophagosome formation in the test pathogen, as confirmed by transmission electron microscopy (TEM) and monodansylcadaverine (MDC) staining. Additionally, a positive correlation was identified between the FeCl_3_ concentration and the damage rate of the fungal sporophyte cell membrane, as the staining rates of the control (untreated), 1/2 MIC, and MIC FeCl_3_ treatment groups were 1.87%, 6.52%, and 18.15%, respectively. Furthermore, the ROS content in sporophyte cells increased by 3.6%, 29.27%, and 52.33% in the control, 1/2 MIC, and MIC FeCl_3_ groups, respectively. Therefore, FeCl_3_ could reduce the virulence and pathogenicity of *C. gloeosporioides*. Finally, FeCl_3_-handled citrus fruit exhibited similar physiological qualities to water-handled fruit. The results show that FeCl_3_ may prove to be a good substitute for the treatment of citrus anthracnose in the future.

## 1. Introduction

Anthracnose caused by *Colletotrichum gloeosporioides* is a tree and fruit disease that seriously affects the quality and postharvest storage of citrus fruit and causes considerable economic losses [1,2]. Although the occurrence of this disease can be controlled through orchard management methods such as plant removal and crop rotation [3], chemical fungicides such as mancozeb and copper compounds, alone or in combination with fosetyl-Al, are the most effective and commonly used agents to manage this disease [4,5,6]. The continuous use of chemical fungicides not only promotes the occurrence of fungicide-resistant pathogenic fungi but also leads to the accumulation of chemical residues in the environment, thus threatening human and ecosystem health [7]. Therefore, developing novel green and efficient alternative methods to manage citrus anthracnose is crucial.

In this study, we investigated the inhibitory effects and mechanisms of FeCl_3_ on *C. gloeosporioides*. Previous studies have found that iron chelates, iron ion complexes, and organic acid iron salts possess excellent antifungal properties and affect a series of growth and development processes [8,9]. Moreover, iron salts are generally considered safe and are commonly used in the food industry. In fact, some iron salts are used to fortify food, as previously reported by the National Health and Family Planning Commission of China, the “Standard for the Use of Food Nutritional Fortifiers” (GB 14880-2012), and the “Standard for the Use of Food Additives” (GB 2760-2014). In addition to its use as a food additive, FeCl_3_ is also used in medicine [10].

Autophagy has been increasingly important in plant–pathogen interactions in recent years [11]. Macromolecular proteins, organelles, ribosomes, and other cell components are degraded by the process of autophagy in response to cellular stress circumstances such as inadequate energy supply, hunger, and environmental stress [12,13]. In addition to influencing and promoting programmed cell death [14], autophagy is essential for a number of regulatory processes, including growth and development [15,16,17,18].

We discovered that FeCl_3_ can promote the autophagy of fungi, reducing the incidence of citrus anthracnose, in the control experiment of citrus anthracnose with FeCl_3_. FeCl_3_’s method of action against *C. gloeosporioides*, however, has not been documented. Through in vitro and in vivo tests, this work investigated the impact of FeCl_3_ solution on the pathogenicity of *C. gloeosporioides* and its suppression on mycelia growth. The analysis of the spore germination rate, cell membrane integrity, autophagy structure, spore activity, and ROS buildup after treatment, as well as the outcomes of storage tests in producing regions, led to a discussion of the potential inhibitory mechanism of FeCl_3_ on *C. gloeosporioides*.

## 2. Materials and Methods

### 2.1. Fungal Pathogens, Citrus, and Metal Salt

*C. gloeosporioides* was donated by professor Yanping Fu from the College of Plant Science & Technology of Huazhong Agricultural University (*C. gloeosporioides* is not currently particularly sensitive or resistant to fungicides). It was cultured on potato dextrose agar (PDA) medium at 28 °C for 7–10 days. ‘Newhall’ navel orange fruits (*C. sinensis* Osbeck) were harvested from the citrus orchard of Huazhong Agricultural University in Wuhan, China, and a commercial orchard in Zhijiang, Hubei, China. FeCl_3_ was purchased from Shanghai Wokai Biotechnology Co., Ltd. (Shanghai, China).

### 2.2. Effect of FeCl_3_ on C. gloeosporioides In Vitro

The antifungal activity of FeCl_3_ on *C. gloeosporioides* was assessed as described by Liu et al. [19] with some modifications. FeCl_3_ was mixed into PDA media to obtain final concentrations of 0.15, 0.3, 0.6, 1.2, and 2.4 g/L. The media were then poured into sterilized 90-mm-diameter Petri dishes. Once the media had cooled and solidified, a 6 mm lawn of fresh *C. gloeosporioides* was placed at the center and incubated at 28 °C for 6–7 d. The mycelial diameter of every plate was recorded via the interior extrapolation method at six days. The lowest concentration that inhibited mycelial growth after two days was defined as the minimum inhibitory concentration (MIC), and the lowest concentration that completely inhibited mycelial growth after four days was defined as the minimum fungicidal concentration (MFC). All experiments were conducted in triplicate.

### 2.3. Effect of FeCl_3_ on C. gloeosporioides In Vivo

Citrus fruits were treated with sodium hypochlorite solution (2%, *v*/*v*) for 2 min, then washed with sterile water and allowed to air dry. As described by Cui et al. [20], a wound (5 mm in diameter and 5 mm in depth) was made around the equator using a punch needle, after which the wound was inoculated with 10 µL of *C. gloeosporioides* spore suspension (1 × 10^6^ spores mL^−1^). After drying for 1–2 min, 10 µL of FeCl_3_ solution at 10× MFC and 15× MFC was added to the wounds, respectively, and sterile water was used as a control. The treated fruit wounds were observed after storing the fruit in a plastic case with water at the bottom at room temperature for a week. Each treatment consisted of 15 fruits, and the experiments were conducted in triplicate.

### 2.4. Effect of FeCl_3_ on C. gloeosporioides Spore Germination

The spore suspensions (1 × 10^6^ spores mL^−1^) of fresh *C. gloeosporioides* were prepared with 1 mL PDB medium in 1.5 mL sterile centrifuge tubes, after which FeCl_3_ was added to the final concentrations of 1/2 MIC, MIC, and MFC, adding sterile water as a control. The suspensions were then cultivated at 28 °C and 180 rpm on a shaker for 12 h, after which the samples were examined with an optical microscope (OM). Each treatment contained 200 spores from at least three replicates, and spore germination rates were calculated as follows: Spore germination = (number of germination spores/total spores) × 100%.

### 2.5. Observation by OM, SEM, and TEM

OM: PDA media containing FeCl_3_ at the MIC and without FeCl_3_ were applied onto cellophane, over which a 6-mm lawn plate of fresh *C. gloeosporioides* was placed. All samples were cultured in an incubator at 28 °C for 6–7 d. Afterward, 5 × 5 mm pieces of cellophane covered with mycelium were cut and observed by OM.

SEM: The pieces of cellophane covered with *C. gloeosporioides* mycelium were prepared following the same method as for OM. However, all sample pieces were fixed in glutaraldehyde solution (3%, *v*/*v*) at 4 °C for 12 h and washed three times with phosphate buffer solution (PBS). Afterward, the samples were dehydrated two times with an ethanol gradient (30%, 50%, 70%, 95%, and 100%, *v*/*v*; 20 min per ethanol solution), after which they were dried at a critical point in liquid CO_2_ and coated with a layer of gold. All samples were observed via SEM (JEOL, JSM-6390LV, Tokyo, Japan).

TEM: The fresh spore solution of *C. gloeosporioides* (1 × 10^4^ conidia/mL) was added into 50 mL PDB medium, then cultivated at 28 °C and 180 rpm in a shaking incubator for 24 h. Next, FeCl_3_ was added to reach a final concentration of 0 and MIC. Afterward, 3–5 mycelium pellets were selected and cultured for 12 h at 28 °C and 180 rpm. The pellets were then fixed in glutaraldehyde solution (3%, *v*/*v*) for 12 h and stored for sample preparation. The samples were postfixed again with osmic acid solution (1%, *v*/*v*) for 2 h. All specimens were washed three times with PBS and dehydrated two times with an ethanol gradient (30%, 50%, 70%, 95%, and 100%, *v*/*v*; 20 min per ethanol solution), followed by polymerization in 21-well silicate embedded plates at 60 °C for 48 h. Finally, 60-nm thin sections were obtained using an Ultratome Leica UC6, after which the slices were stained in uranyl acetate (2%) and lead citrate for 30 min and 10 min, respectively. Finally, the samples were observed via TEM (Hitachi H-7650, Tokyo, Japan).

### 2.6. Detection of FeCl_3_ on the Cell Membrane Integrity of C. gloeosporioides

PI staining: Fresh *C. gloeosporioides* spore suspension (1 × 10^6^ spore mL^−1^) was prepared with 1 mL PDB staining medium in 1.5 mL sterile centrifuge tubes. Then, it was cultured at 28 °C and 180 rpm on a shaker for 3 h, centrifuged for 3 min at 4000 rpm and washed three times with PBS. The spores were collected and dyed with 200 mL propidium iodide (PI) (Coolaber Technology Co., Ltd., Beijing, China). All samples were stained at 37 °C for 5–10 min, after which they were rinsed twice with PBS to remove excess dye. The samples were then observed under a fluorescence microscope (Nikon Eclipse 90i).

### 2.7. Detection of Autophagic Structures, Conidial Viability and ROS Accumulation

A total of 1 mL of *C. gloeosporioides* conidial suspension (1 × 10^6^ conidia/mL) was prepared, and FeCl_3_ was added to reach a final concentration of 1/2 MIC and MIC, followed by incubation at 28 °C and 180 rpm on a shaker for 3 h before staining. Conidial viability was detected using fluorescein diacetate (FDA, 5 g L^−1^). The samples were then incubated with the corresponding stains in the dark for 5 min, after which they were rinsed twice with PBS (pH = 7.0). The autophagosomes and intracellular ROS accumulation of the conidia were detected using a monodansylcadaverine assay kit (MDC, Solarbio, Beijing, China) and a Reactive Oxygen Species (ROS) assay kit (Solarbio, Beijing). All samples were observed under a fluorescence microscope (Nikon Eclipse 90i).

### 2.8. Physiological Qualities Test of FeCl_3_-Treated Citrus Fruit

The physiological qualities of FeCl_3_-treated citrus fruit were detected. For 30–45 s, ‘Newhall’ navel orange fruits (*C. sinensis* Osbeck) were immersed in 10× MFC FeCl_3_ (=24 g/L), water, and chemicals. All fruits were transferred into a climate chamber for 30 days of storage. Total soluble solids (TSS) and titratable acid (TA) were tested using a Pocket Brix acidity meter (ATAGO Co., Ltd., Saitama, Japan) in accordance with the operation manual; 2,6-dichlorophenolindophenol was used to determine the VC content in fruit, there were five fruits in each group, and trials were repeated three times. Fruit weight losses were measured using an electronic scale, there were ten fruits in each group, and trials were repeated three times.

### 2.9. Statistical Analysis

All experiments were conducted in triplicate following a completely randomized design. Significant differences were determined via one-way ANOVA followed by Duncan’s multiple range test (SPSS 26.0, *p* < 0.05).

## 3. Results

### 3.1. Inhibiting Effect In Vitro and In Vivo

The in vitro growth of *C. gloeosporioides* mycelia treated with FeCl_3_ was significantly inhibited in the PDA medium (Figure 1A,B). Without FeCl_3_ treatment, the colony center became dense and thickened, and the aging hyphae exhibited a greyish green coloration. This color disappeared when the FeCl_3_ concentration reached 1.2 g/L, and mycelium growth was significantly inhibited. When the FeCl_3_ concentration reached 2.4 g/L, colony growth was completely inhibited.

As illustrated in Figure 1C, the in vivo antifungal activity of FeCl_3_ increased in a dose-dependent manner. When the concentration of FeCl_3_ in citrus reached 15× MFC, the number of *C. gloeosporioides* cells in the treatment group was significantly lower than that in the control group after one week of inoculating the spore suspension.

### 3.2. Effect of FeCl_3_ on the Spore Germination

The spore germination of *C. gloeosporioides* was remarkably inhibited by FeCl_3_ (Figure 2A). When the FeCl_3_ concentration reached 1/2 MIC and MIC, the germination rates of *C. gloeosporioides* were 55.77% and 4.83%, respectively, which represented a significant decrease compared with the control (88.23%, *p* < 0.05) (Figure 2B). Moreover, the length of the germ tubes in the FeCl_3_-treated group was markedly shorter than that in the control group.

### 3.3. Effect of FeCl_3_ on Mycelial Morphology and the Cell Membrane Integrity

OM and SEM: As observed by OM and SEM, normal hyphae exhibited a homogeneous and linear smooth surface, whereas those treated with MIC FeCl_3_ were severely damaged, with irregularly contracted or expanded cell membranes, or were even bound together (marked with red arrows) (Figure 3A).

PI staining: As shown by PI staining, the number of inactivated spores increased as the concentration of FeCl_3_ increased. Almost all spores (98.13%) in the control group were not stained (Figure 3B), whereas substantially more *C. gloeosporioides* spores (6.52% and 18.15%) were stained in the 1/2 MIC and MIC treatment groups (Figure 3C), indicating that the membrane of the *C. gloeosporioides* spores was damaged.

### 3.4. Effect of FeCl_3_ on Internal Structure and Autophagic Activity

Surprisingly, our TEM observations revealed the presence of transparent irregular-shaped vacuolar structures in the cytosol of the FeCl_3_-treated conidia (Figure 4A). Therefore, MDC staining was conducted to identify these structures. As shown in Figure 4B, the FeCl_3_-treated spores exhibited concentrated spots of MDC fluorescence, whereas the normal spores were uniformly dyed in MDC fluorescence. Apoptotic spores were also observed and appeared as stained particles of varying sizes and densities, indicating that their chromatin condensed, and the cell nucleus burst. Furthermore, these effects were more notable at higher FeCl_3_ concentrations (as shown by the arrow).

### 3.5. Effect of FeCl_3_ on Vital Activity and ROS Accumulation

Overall, spores with high viability were detected by FDA green fluorescence. As shown in Figure 5A, there was no significant difference between the stained spores of the control and treated groups, indicating that FeCl_3_ did not inactivate the spores. However, in the detection of reactive oxygen species, the control spores (3.61%) were rarely detected with high DCHF-DA fluorescence intensity labeling, whereas *C. gloeosporioides* spores cultured with 1/2 MIC and MIC FeCl_3_ showed 29.27% and 52.33% staining, respectively (Figure 5B,C). This suggests that FeCl_3_ induces the accumulation of ROS within the spores of *C. gloeosporioides*.

### 3.6. Physiological Qualities Test of FeCl_3_-Treated Citrus Fruit

After 30 days of storage, citrus fruit treated with FeCl_3_ displayed TSS, TA, and VC values that were comparable to those of citrus fruit treated with water and a chemical fungicide (Figure 6A–C). These three groups all saw fruit weight decreases of roughly 2.5%. (Figure 6D). These findings demonstrated that FeCl_3_ had no byproduct effects on citrus fruit.

## 4. Discussion

Anthracnose is one of the most prevalent diseases among a variety of fruit trees, particularly citrus, apple, mango, and other fruit trees [21,22,23]. As discussed above, mancozeb and copper compounds alone or coupled with fosetyl-Al are chemical fungicides that are commonly used in agriculture [4,5,6]. However, the uncontrolled use of these agents can promote the occurrence of fungicide-resistant pathogenic fungi, in addition to affecting human and environmental health [7,24]. Autophagy research has mainly focused on basic biology, whereas relatively few studies have been conducted on plants and plant pathogens. Nevertheless, the study of plant antibacterial mechanisms has recently garnered increasing attention [25]. Our studies demonstrated that autophagy could be exploited for the development of green disease prevention and treatment strategies in croplands.

TEM and MDC fluorescence staining are two commonly used methods to confirm the occurrence of autophagy [26]. In this study, our TEM and MDC fluorescence staining observations confirmed the formation of autophagy structures in FeCl_3_-treated *C. gloeosporioides* spores (Figure 4A). Numerous studies have demonstrated that autophagy plays a crucial role in pathogen–plant interaction [27], which is consistent with the findings of this study.

Autophagy is rarely studied in fungi, and most of the research data are focused on biomedicine and plants. It has been reported that autophagy plays an important role in the life cycle of fungi, and most autophagy is beneficial to life itself. However, this viewpoint has not been reported for *C. gloeosporioides*. According to the experimental results in vitro and in vivo, FeCl_3_-treated *C. gloeosporioides* not only produce autophagy phenomena but also have their growth significantly inhibited. Therefore, it is reasonable to assume that FeCl_3_-induced autophagy of *C. gloeosporioides* is harmful to *C. gloeosporioides*.

The FeCl_3_-treated *C. gloeosporioides* spores exhibited significant growth inhibition both in vitro and in vivo, suggesting that FeCl_3_ exerted a potent antifungal effect (Figure 2). The results of the spore germination experiment showed that the germination rate of spores in the FeCl_3_ treatment group decreased with higher FeCl_3_ concentrations. Moreover, the diameter and length of the germinated spores in the treated group were smaller than in the control group. Therefore, MDC fluorescence staining was used to observe the internal changes of the spore cells (Figure 3B). SEM and TEM were used to observe the external morphological changes of hyphae and the changes in the spore cells of the treated groups (Figure 3A and Figure 4A). Our findings demonstrated that, unlike the control group, the chromatin of the FeCl_3_-treated spore cells was condensed, as demonstrated by the occurrence of concentrated bright spots in the MDC fluorescence staining experiments. In the control group, the staining was uniformly spread throughout the spore cells. Additionally, our SEM observations revealed that the hyphae of pathogenic fungi exhibited different degrees of coiling and shrinking. Our TEM observations revealed the occurrence of various double-membrane vesicular autophagic structures and countless vacuolated structures in the interior of the spore cells. Our findings suggest that the generation of autophagy structures affects spore germination and the formation of infective structures of pathogenic fungi, thereby reducing the invasiveness and pathogenicity of pathogenic fungi (Figure 7). Previous studies have demonstrated that autophagy can affect the growth, vitality, and pathogenicity of pathogenic fungi by inducing autophagy-related functions and apoptosis [20,28]. Therefore, we concluded that FeCl_3_ triggers apoptosis by inducing autophagy in *C. gloeosporioides*.

Interestingly, our PI and FDA staining experiments demonstrated that although the FeCl_3_-treated spores exhibited some degree of membrane damage, the cells did not die and were still viable (Figure 3B and Figure 5A). These results were inconsistent with our original hypothesis that the induction of autophagy inhibits hyphal growth and triggers cell apoptosis in pathogenic fungi. Moreover, previous reports have demonstrated that autophagy promotes cell apoptosis in pathogenic fungi, which is also inconsistent with our findings [29]. After studying the inhibition mechanism of iron ion magnetic nanomaterials on model fungi, Qi Peng et al. concluded that the antifungal properties of iron ions could not be attributed to common plasma membrane damage or cell wall damage [30]. Therefore, we performed DCFH-DA fluorescent staining on the spores of the pathogenic fungi (Figure 5B,C). In the FeCl_3_ treatment group, the DCFH-DA staining rate increased at higher FeCl_3_ concentrations. Furthermore, our TEM observations revealed the occurrence of irreversible degradation of some organelles in the spore cells, such as mitochondria. The formation of autophagic structures leads to the degradation of the contents of the spore cells, which includes but is not limited to mitochondria.

Combining the results of the above in vitro and in vivo experiments, we found that FeCl_3_ significantly inhibited the growth of *C. goleosporioides*, both in mycelium growth and spore germination. Zhonghuan Tian et al. explored the physiological quality of fruits in terms of their application in producing areas [31]. In order to explore the application performance of FeCl_3_ in actual production, we conducted a post-harvest storage and fresh-keeping experiment on ‘Newhall’ navel orange fruits (*C. sinensis* Osbeck) in a citrus industry orchard in Zhijiang (Hubei, China). The post-harvest treatment of fruit is simulated by soaking the fruit. After one month of positive storage, the results showed that the total soluble solid (TSS), titrable acid (TA), L-ascorbic acid (VC), and fruit weight loss were not significantly affected by water, FeCl_3_, and chemical treatments. These results suggest that FeCl_3_ has no by-product effect on citrus fruits [32]. In this regard, we believe that FeCl_3_ can be used as a substitute for chemical fungicides to control anthracnose in producing areas in the future.

Therefore, our findings provided insights into the inhibitory mechanisms of FeCl_3_ on *C. gloeosporioides* (Figure 7). Here, we conclude that FeCl_3_ could induce autophagy for self-protection and increase the ROS content in pathogenic fungi. FeCl_3_ could damage the internal structure of pathogenic fungi and induce the abnormal development of pathogenic fungi infection structures, thus decreasing the pathogenicity and virulence of *C. gloeosporioides*. Further assays are required to determine the controlling effect of anthracnose during citrus storage under general agronomic situations when FeCl_3_ is applied, and research to detect its potential compatibility with other treatments is desired. FeCl_3_, a type of iron salt with low cost and safety, offers a new option for replacing existing chemical fungicides.

## 5. Conclusions

Our findings demonstrate that FeCl_3_ effectively inhibited the growth of *C. gloeosporioides* both in vitro and in vivo. FeCl_3_ induced autophagosome formation and ROS accumulation, leading to organelle destruction and increased cell membrane permeability; however, it did not cause lethality. Our preliminary analysis of the mechanism of action and targets of FeCl_3_ provides a theoretical basis for the future application of FeCl_3_ to control citrus anthracnose. Additionally, our findings could serve as a guide for the future elucidation and characterization of invasion-related genes. However, additional studies are still needed to gain a more in-depth understanding of the molecular mechanisms through which FeCl_3_ inhibits the growth of *C. gloeosporioides*. Based on the experiments in Zhijiang (Hubei, China), FeCl_3_ is expected to replace chemical agents in the control of anthracnose in the future.

## Figures and Tables

**Figure 1 jof-09-00230-f001:**
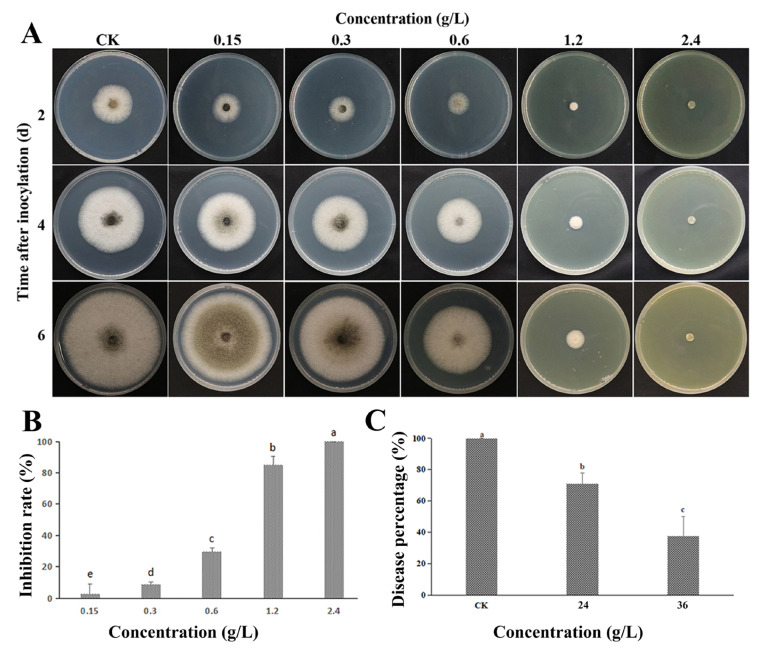
Inhibitory effect of FeCl_3_ on *C. gloeosporioides* growth in vitro and in vivo. (**A**) FeCl_3_ inhibits the mycelial growth of *C. gloeosporioides* in vitro; (**B**) Statistical analysis of the effects of different FeCl_3_ concentrations on the mycelial growth of *C. gloeosporioides* (6 d); (**C**) FeCl_3_ inhibits *C. gloeosporioides* virulence in vivo (7 d); the control group (CK) was treated with sterile water. All data are reported as the mean ± SD. Different letters indicate significant differences according to Duncan’s multiple range test (*p* < 0.05).

**Figure 2 jof-09-00230-f002:**
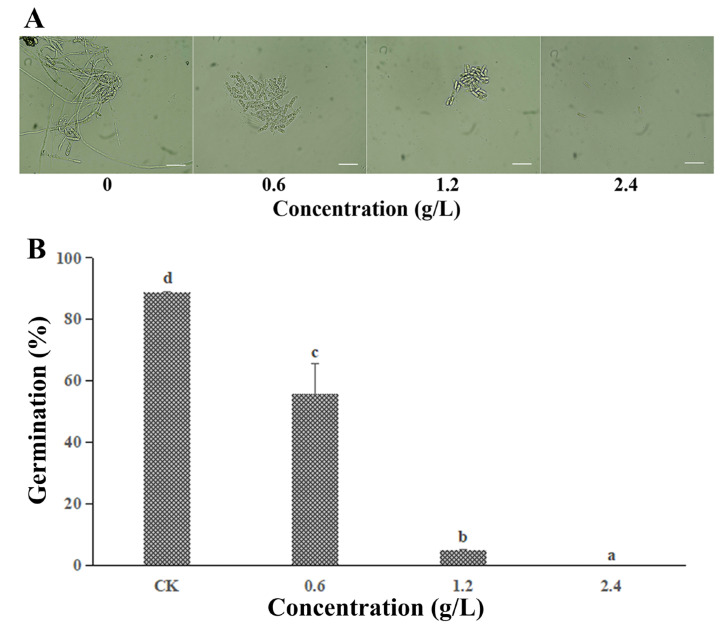
Effects of FeCl_3_ treatments on *C. gloeosporioides* spore germination. (**A**) Inhibition efficacy of FeCl_3_ on *C. gloeosporioides* spore germination (scale bar = 20 µm); (**B**) Spore germination rates. All data are reported as the mean ± SD. Different letters indicate significant differences according to Duncan’s multiple range test (*p* < 0.05).

**Figure 3 jof-09-00230-f003:**
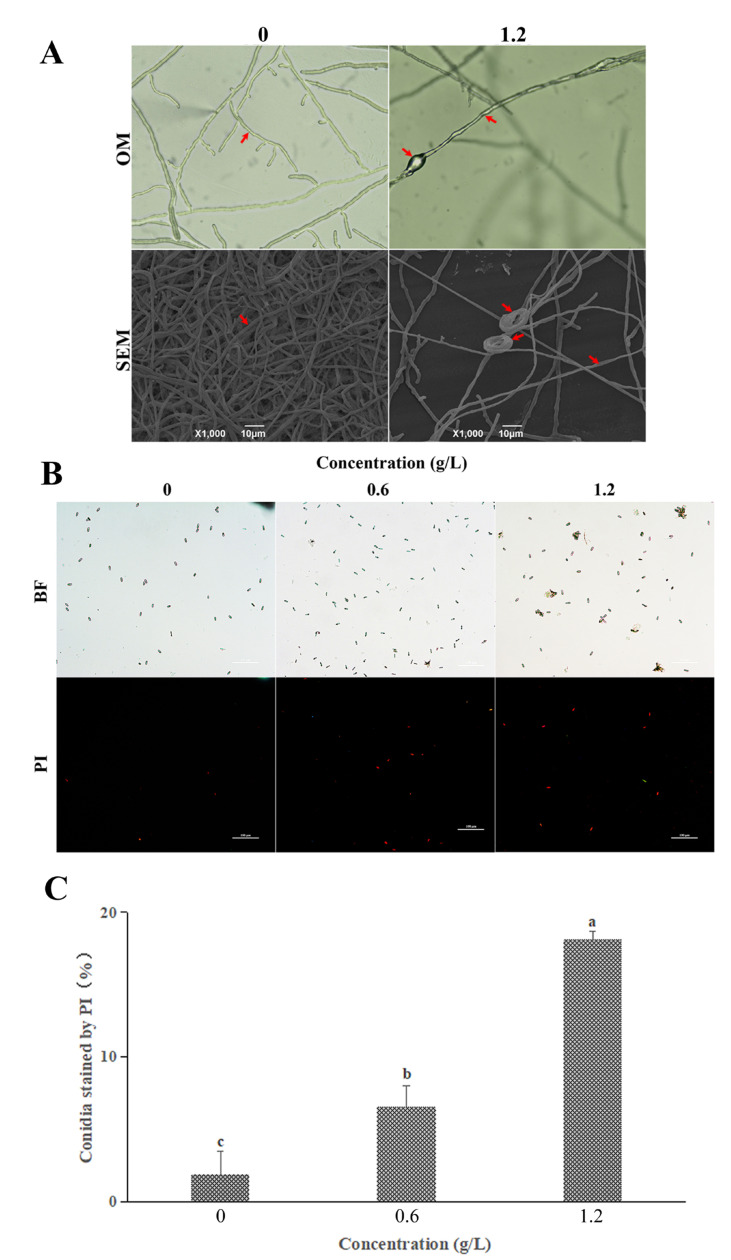
Effects of FeCl_3_ on the cell membrane integrity of *C. gloeosporioides*. (**A**) Observation of mycelium morphology by OM and SEM (hyphal shrinkage, depression, coiling, and winding are indicated by arrows); (**B**) Observation of PI staining; (**C**) Statistical analysis for spores stained with PI. The error bars indicate the standard error of the mean of three replicate isolates. All data are reported as the mean ± SD. Different letters indicate significant differences according to Duncan’s multiple range test (*p* < 0.05).

**Figure 4 jof-09-00230-f004:**
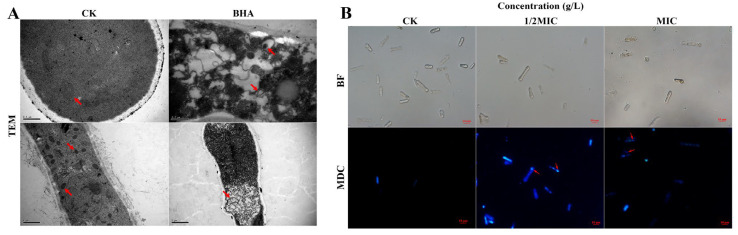
FeCl_3_ triggers autophagosome formation. (**A**) Observation of internal structures by TEM (the arrows indicate the double-membrane vesicular structures (autophagosomes)); (**B**) MDC staining of *C. gloeosporioides* (the arrows indicate the condensed chromatin, i.e., autophagosomes).

**Figure 5 jof-09-00230-f005:**
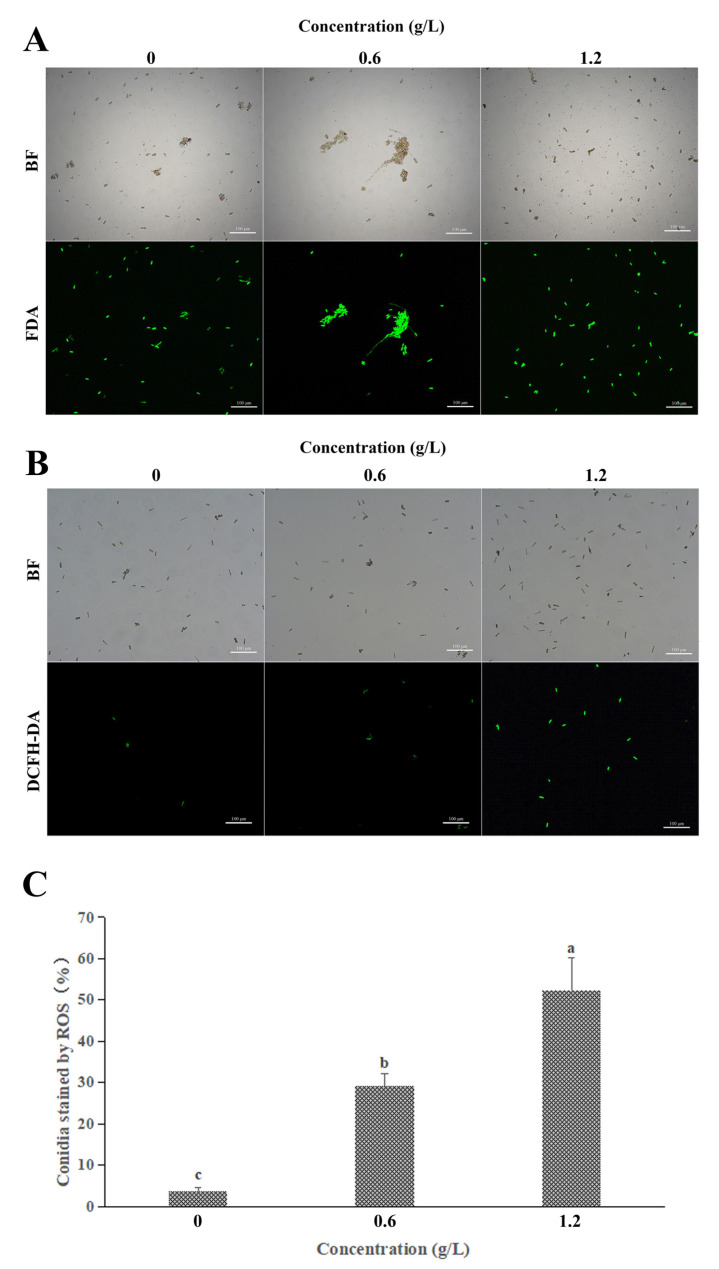
FeCl_3_ decreases cell viability and increases intracellular ROS levels. (**A**) *C. gloeosporioides* conidia stained by FDA; (**B**) *C. gloeosporioides* conidia stained by DCFH-DA; (**C**) Statistical analysis of excessive ROS accumulation. All data are reported as the mean ± SD. Different letters indicate significant differences according to Duncan’s multiple range test (*p* < 0.05).

**Figure 6 jof-09-00230-f006:**
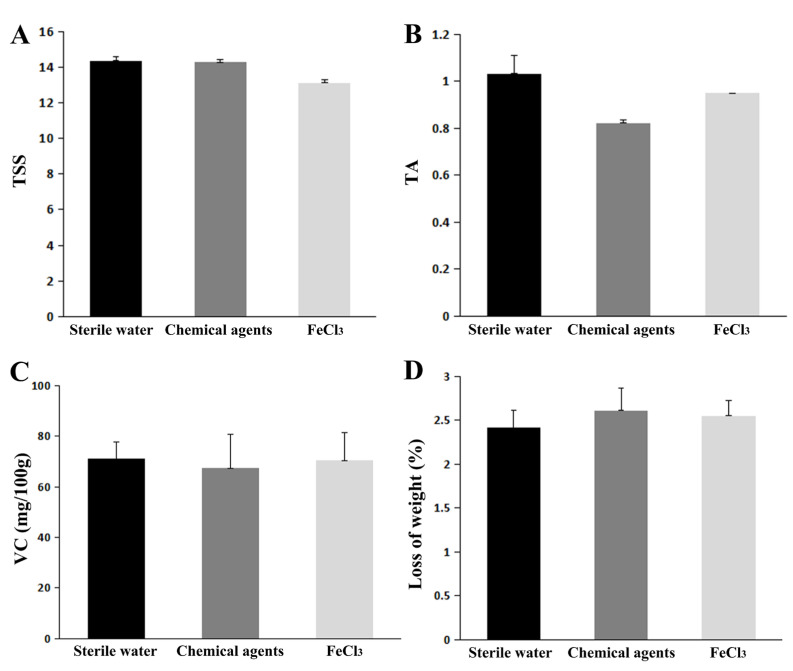
Physiological qualities test of FeCl_3_-treated citrus fruit. (**A**) Total soluble solids (TSS). (**B**) Titratable acid (TA). (**C**) L-Ascorbic acid (VC). (**D**) Fruit weight loss. The fruits were stored in the ventilating chamber for 30 days.

**Figure 7 jof-09-00230-f007:**
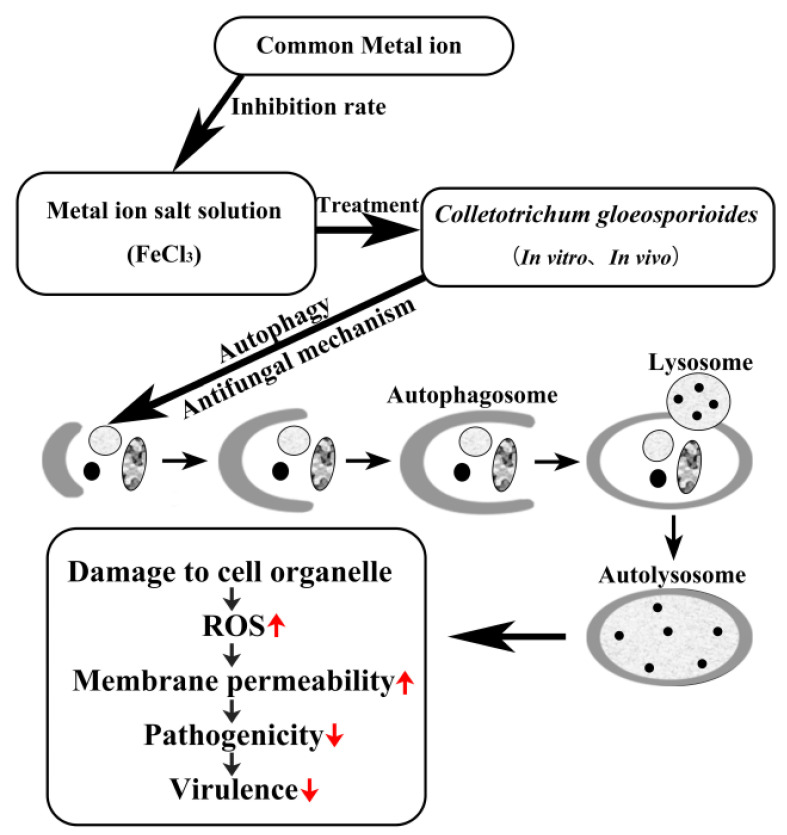
Hypothetic model of the effect of FeCl_3_ treatment on *C. gloeosporioides* pathogenicity. ↑: up-regulated; ↓: down-regulated.

## Data Availability

No new data were created in this study. Data sharing is not applicable to this article.

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
