# Peer review of "Ferric Chloride Controls Citrus Anthracnose by Inducing the Autophagy Activity of Colletotrichum gloeosporioides"

_jof, 2023, doi:10.3390/jof9020230_

Round 1

Reviewer 1 Report

The work entitled Ferric chloride control citrus anthracnose by inducing the autophagy activity of Colletotrichum gloeosporioides try to figure out the mechanism of the inhibitory effect of ferric chloride against C. gloeosporioides.

 However, numerous deficiencies have been detected in the wording and in the presentation of the results that are detailed below.

 Please use italics in all terms in vivo or in vitro

 Line 67-69: This phrase is not clear, in previous experiments, in this work? Do not include results in the introduction, please redo the sentence.

 The characteristics of the C. gloeosporioides isolate used are not described. It is not known if it is a sensitive or resistant isolate to fungicides.

 Line 92: The mycelial diameter of every plate was record base on interior extrapolation method for every day-Please revise

Line 103,110,130,153: review all exponential forms in spore concentrations (106)

 Figure 1 specify what CK is. It is not indicated at what time the percentage is calculated 2, 4 or 6 days? the x-axis does not specify concentration of what? Once again, section C does not specify at what time the infection percentage is calculated. An image of the infected fruits is missing. Why hasn't the incidence or severity been also shown in the form of the diameter of the infection? Revise all figure legends

In figure 2 what is the difference between the top and bottom of panel A? The image quality is not good and the differences are not clearly perceived.

 The quality of figure 3 is not good. I don't understand the red arrows on the control. It is very difficult to see the differences in PI staining. concentrations. In panel C the concentrations are equal to 1/2 MIC and MIC? The percentages are much higher than those that are detected in plain sight

In figure 4 again the quality of the images is not good and it is very difficult to detect what the authors show.

To ensure what is said in the section Effect of ferric chloride on vital activity and ROS accumulation, the quality of the images needs to be greatly improved. I cannot be able to conclude what they show.

Since many of the results are based on the images and these are not of quality, it is difficult to assess the discussion and the conclusions reached by the authors.

Reviewer 2 Report

It was interesting to read the findings of this study considering the importance of Colletotrichum as a major plant pathogen. While iron oxide nanoparticles from Mentha spicata were reported to inhibit Phytophthora infestans (Khan et al 2022 Frontiers in Plant Science) and Manandhar et al (1998) report ferric chloride for management of blast of rice.

Autophagy is “consumption of the body’s own tissue as a metabolic process occurring in starvation and certain diseases”. In other words, it is the natural, conserved degradation of the cell that removes unnecessary or dysfunctional components through a lysosome-dependent regulated mechanism. In fact, often it is considered as a beneficial mechanism. Extensive studies on functions of autophagy-related genes have revealed that autophagy plays a role in cell differentiation and pathogenesis of pathogenic fungi (Jiang et al 2020.). Autophagy related genes (ATGs) are essential for growth, development and sexual reproduction (Zhao et al., 2019). Wang et al (2017): Genome-wide functional analysis reveals that autophagy is necessary for growth, sporulation, deoxynivalenol production and virulence in Fusarium graminearum.

·         So, the question is, are we helping the pathogen by inducing autophagy. The authors also claim that ferric chloride could reduce the virulence and pathogenicity of C. gloeosporioides.

·         The second main concern is entire study is based on some in vitro trials. Unless these findings are field-validated, we cannot claim that ferric chloride could be recommended for management of Colletotrichum and the results get confined to journal.

·         The authors could as well abbreviate ferric chloride subsequently.

·         The review is inadequate as autophagy is being extensively studied and the gap analysis for present study is not properly done.

·         The language needs a very thorough revisit as there are many technical as well as grammatical mistakes.

Considering above aspects, I suggest the paper may be resubmitted with additional results. Definitely, it would be disheartening to the authors, but the observations are for enhanced valorization of the research findings.

Round 2

Reviewer 1 Report

After evaluating the modifications included by the authors following the reviewer's instructions, the quality of the manuscript has improved to be accepted for publication.

Reviewer 2 Report

The authors have responded to all queries but for the first query as to whether the pathogen is benefitted by the authophagy induced over time.

In the newly added text, there are minor language mistakes and need attention.

IF the Editor-in-Chief is convicned with the explanation, it may be accepted.
